# Electroreductive hydroxy fluorosulfonylation of alkenes

Qingyuan Feng[1], Tianyu He[1], Shencheng Qian [1], Peng Xu[1], Saihu Liao [2] & Shenlin Huang [1] ✉

An electroreductive strategy for radical hydroxyl fluorosulfonylation of alkenes with sulfuryl chlorofluoride and molecular oxygen from air is described. This mild protocol displays excellent functional group compatibility, broad scope, and good scalability, providing convenient access to diverse β-hydroxy sulfonyl fluorides. These β-hydroxy sulfonyl fluoride products can be further converted to valuable aliphatic sulfonyl fluorides, β-keto sulfonyl fluorides, and β-alkenyl sulfonyl fluorides. Further, some of these products showed excellent inhibitory activity against *Botrytis cinerea* or *Bursaphelenchus xylophilus,* which could be useful for potent agrochemical discovery. Preliminary mechanistic studies indicate that this transformation is achieved through rapid $O_2$ interception by the alkyl radical and subsequent reduction of the peroxy radical, which outcompete other side reactions such as chlorine atom transfer, hydrogen atom transfer, and Russell fragmentation.

Sulfonyl fluorides have found wide applications in chemical biology[1,2], materials science[3,4], organic synthesis[5,6], and other areas in chemistry[7,8], since sulfur(VI) fluoride exchange (SuFEx) reactions were recognized as a new generation of click reaction in 2014[9]. In this context, a range of key methods have been developed for the construction of $FSO_2$-containing molecules[5,7–9], primarily including aryl sulfonyl fluorides[10–13], alkenylsulfonyl fluorides[14–16], alkynylsulfonyl fluorides[17], and β-keto sulfonyl fluorides[18–20]. On the other hand, alcohols are one of the most ubiquitous functional groups in natural products and bioactive molecules. The incorporation of a hydroxyl group can significantly change the binding affinity and pharmacokinetic properties of drug molecules[21–23]. As such, we envisioned that the combined β-hydroxy sulfonyl fluoride motif might exhibit improved bioactivity compared to previously reported β-keto sulfonyl fluorides (Fig. 1a, I)[19].

Radical difunctionalization of alkenes would be an ideal strategy for the construction of β-hydroxy sulfonyl fluoride scaffolds (Fig. 1a., II), as it could allow the simultaneous introduction of HO and $FSO_2$ functionalities onto prevalent alkene feedstocks. Moreover, the direct reduction of the ketone group in β-keto sulfonyl fluorides failed to afford β-hydroxy sulfonyl fluorides in our hands

(Fig. 1b). While several elegant protocols for the radical fluorosulfonylation[24–30] and radical hydroxysulfonylation[31–33] have been developed, 1,2-hydroxy fluorosulfonylation of alkenes remains a synthetic challenge and $FSO_2$ radical precursors are strictly limited so far (Fig. 1a). Recently, Liao discovered that radical **Int-1** abstracted Cl-atom too fast to be trapped with other reagents, and alkenylsulfonyl fluorides were formed from the radical fluorosulfonylation of olefins (Fig. 1c)[29]. Later, the same group avoided the fast Cl-atom transfer process with a benzimidazolium-based sulfonyl fluoride reagent as the radical precursor, enabling the trapping of benzylic carbocation with simple alcohols (Fig. 1c)[27]. Both Studer[28] and Glorius[24] groups introduce bifunctional reagents for the radical olefin 1,2-difunctionalization, providing β-alkynyl sulfonyl fluorides and β-imino sulfonyl fluoride, respectively. Despite these advances, synthetic access to β-hydroxy sulfonyl fluorides has not been reported to date.

Electrochemistry has emerged as a sustainable tool in organic chemistry[34–39], which in many cases is complementary to photoredox catalysis[40]. Indeed, the electrochemical oxidation has been extensively applied for the difunctionalization of alkenes[41–44]. However, the electroreductive strategy[45–50] for the alkene

[1]Jiangsu Co-Innovation Center of Efficient Processing and Utilization of Forest Resources, International Innovation Center for Forest Chemicals and Materials, Nanjing Forestry University, Nanjing 210037, China. [2]State Key Laboratory of Physical Chemistry of Solid Surfaces, Xiamen University, Xiamen 361005, China. ✉e-mail: shuang@njfu.edu.cn

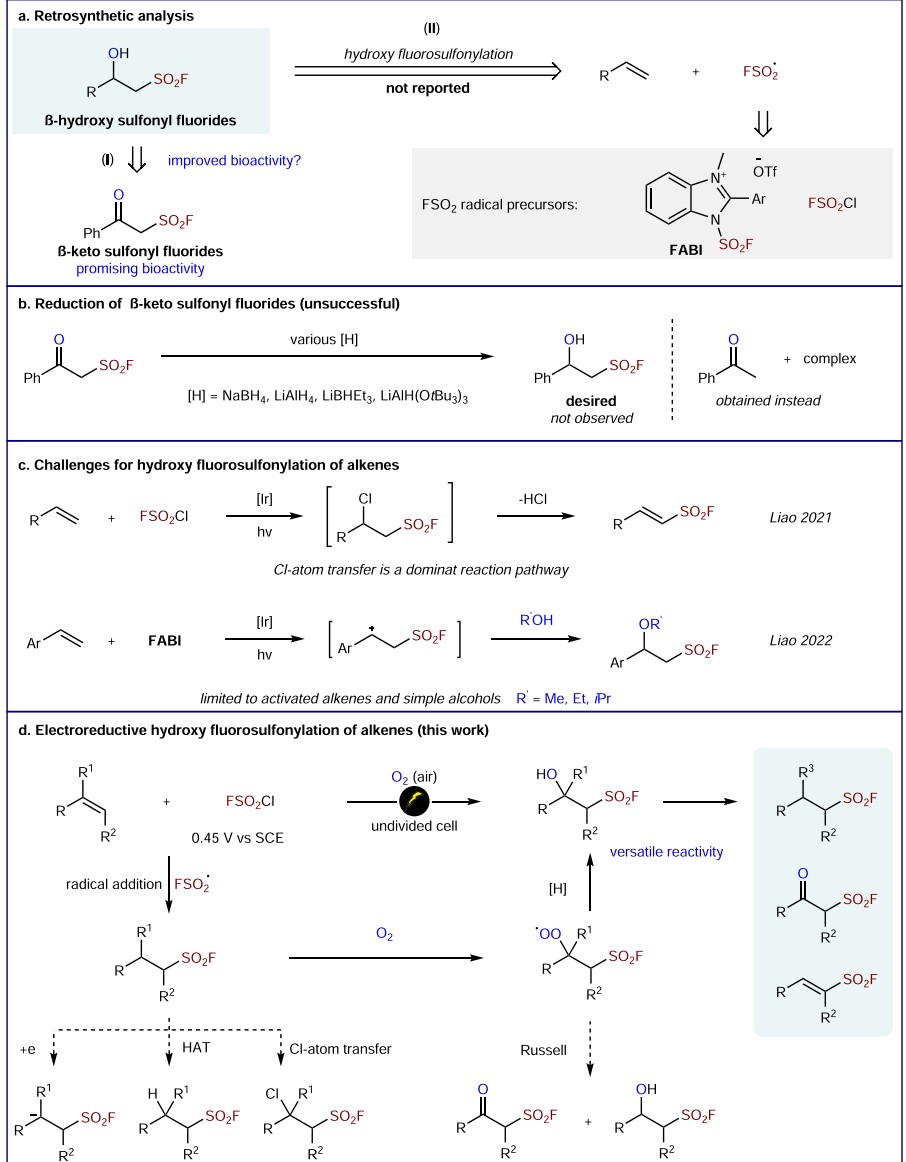

**Fig. 1 | Context of the work. a** Retrosynthetic analysis. **b** Reduction of β-keto sulfonyl fluorides. **c** Challenges for hydroxyl fuorosulfonylation of alkenes. **d** Electroreductive hydroxy fluorosulfonylation of alkenes.

difunctionalizations is substantially less explored. It is noteworthy that an electroreductive radical-polar crossover strategy for the difunctionalization of alkenes has recently been disclosed by Lin[51]. Afterward, several methods for difunctionalization of alkenes were developed rapidly via electroreductive chemistry[52–54]. We herein report the successful development of an electroreductive 1,2-hydroxyl fluorosulfonylation of alkenes, which produces diverse β-hydroxy sulfonyl fluorides (Fig. 1d). Taking advantage of feasible access to FSO₂• at a very low reduction potential (FSO₂Cl, $E_{p/2}$ = 0.45 V vs SCE, see Supplementary Fig. 11), we could avoid the further reduction of alkyl radical intermediate (i.e., benzyl radical, $E_{p/2}$ = −1.6 V vs SCE)[51]. The consumption of FSO₂Cl at the cathode results in a low local concentration of FSO₂Cl. Thus, a rapid O₂ interception by the alkyl radical would outcompete the Cl-atom transfer from the low concentration of FSO₂Cl. Identification of a suitable reduction system would be key to suppressing the competing pathway including Russell fragmentation[55,56], hydrogen atom transfer (HAT)[26], and carbon anion generation. Furthermore, the synthetic utilities have been demonstrated by versatile follow-up derivatizations and biological activity studies.

## Results
### Reaction development

In our initial survey, we investigated the designed 1,2-hydroxyl fluorosulfonylation toward **3** by employing our previous conditions for electrochemical oxo-fluorosulfonylation of phenylacetylene (Fig. 2a)[19]. However, the reaction of styrene (**1a**) and FSO₂Cl (**2**) provided the desired β-hydroxy sulfonyl fluoride **3** in only 10% yield, along with 13% yield of β-keto sulfonyl fluoride **4** and a complex mixture of other inseparable products (Fig. 2b). This result clearly indicates that Russell fragmentation of alkyl peroxy radical may compete with the desired reduction pathway[55,56].

Next, we explored the reaction conditions with styrene (**1a**) and FSO₂Cl (**2**) in an undivided cell equipped with an aluminum plate anode and a zinc plate cathode under air at room temperature (Table 1). When constant current conditions were employed, the potential gradually increased over the reaction time, and more byproducts were formed. As such, constant cell voltage conditions were employed to avoid undesired redox processes. Inspired by Mukaiyama hydration[57,58], various hydride donors were examined to suppress ketone formation. After extensive optimization, we were

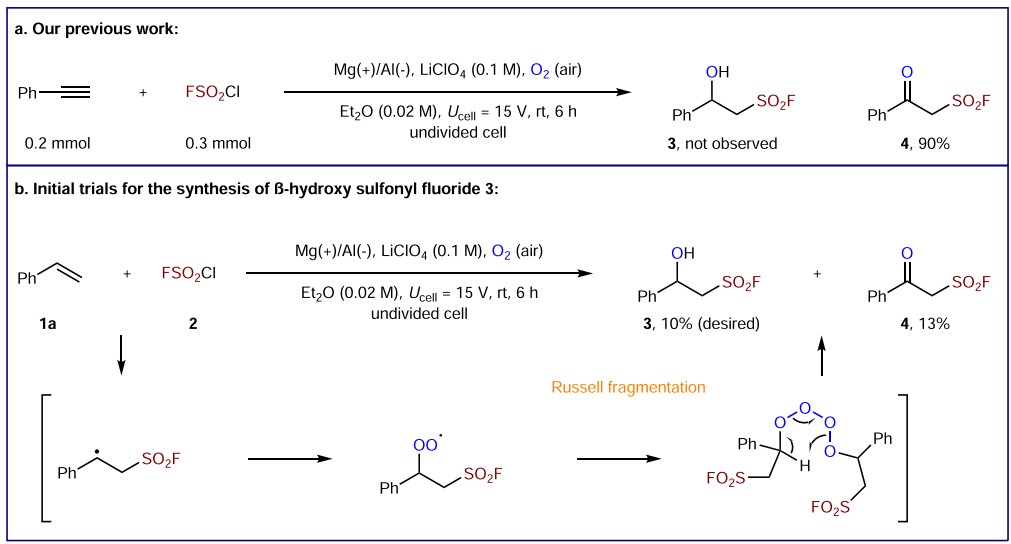

**Fig. 2 | Initial studies for hydroxy fluorosulfonylation. a** Our previous work. **b** Initial trials for the synthesis of β-hydroxy sulfonyl fluoride **3**.

pleased to find that with $Et_3SiH$ (2.0 equiv) and $B_2(OH)_4$ (2.5 equiv) using $LiClO_4$ as the electrolyte in $Et_2O$ (0.016 M) under 8 V constant cell voltage conditions, the desired olefin difunctionalization product **3** was isolated in 96% yield (entry 1). Without $Et_3SiH$ and $B_2(OH)_4$, only 40% yield of **3** was obtained along with other inseparable byproducts (entry 2). Control experiments showed that $Et_3SiH$ and $B_2(OH)_4$ both were important (entries 3&4). Other silanes were screened but resulted in lower reaction efficiency (entries 5–7). The difunctionalization reaction did not proceed when using radical initiator $Et_3B$ or reducing reagent $BH_3·THF$ instead of $B_2(OH)_4$ (entry 8). These results indicated that $B_2(OH)_4$ was not likely employed as a radical initiator or simple

### Table 1 | Optimization of the reaction conditions[a]

| Entry | Variation from standard condition | Yield[b] |
|---|---|---|
| 1 | None | 96% |
| 2 | Without $Et_3SiH$ and $B_2(OH)_4$ | 40% |
| 3 | Without $B_2(OH)_4$ | 69% |
| 4 | Without $Et_3SiH$ | 74% |
| 5 | $PhSiH_3$ instead of $Et_3SiH$ | 40% |
| 6 | $(EtO)_3SiH$ instead of $Et_3SiH$ | 51% |
| 7 | $tBuMe_2SiH$ instead of $Et_3SiH$ | 70% |
| 8 | $Et_3B$ or $BH_3·THF$ instead of $B_2(OH)_4$ | 0% |
| 9 | $BF_3·Et_2O$ instead of $B_2(OH)_4$ | 60% |
| 10 | 20 mol% $B(C_6F_5)_3$ instead of $B_2(OH)_4$ | 76% |
| 11 | Zn(+)/Al(−) instead of Al(+)/Zn(−) | 32% |
| 12 | Al(+)/Al(−) instead of Al(+)/Zn(−) | 38% |
| 13 | Zn(+)/Zn(−) or C(+)/C(-) | 0% |
| 14 | GF(+)/Zn(−) | 75% |
| 15 | THF as the solvent | 27% |
| 16 | 1,4-dioxane as the solvent | 49% |
| 17 | $CH_2Cl_2$, MeCN, HFIP, or TFE | 0% |
| 18 | $Et_2O$ (0.02 M) | 77% |
| 19 | $Et_2O$ (0.013 M) | 73% |
| 20 | Under $N_2$ | 0% |
| 21 | No electricity | 78% |

[a]Conditions: **1a** (0.2 mmol), **2** (2 equiv), $Et_3SiH$ (2 equiv), $B_2(OH)_4$ (2.5 equiv), $LiClO_4$ (0.1 M), $Et_2O$ (0.016 M), aluminum anode (10 mm × 15 mm × 1 mm), Zinc cathode (10 mm × 15 mm × 1 mm), cell voltage ($U_{cell}$ = 8 V), undivided cell, 0.89 F/mol, air, rt, 16 h. [b]Isolated yield.

borane precursor. Lewis acids $BF_3·Et_2O$ and $B(C_6F_5)_3$ led to decreased yields (entries 9 and 10). Since $B_2(OH)_4$ could be used as a deoxygenating agent[59] and boronic acids were able to reduce peroxides[60], here $B_2(OH)_4$ likely acted as a reducing agent for the reduction of the hydroperoxide intermediate. It is known that the electrode material can significantly influence electron transfer[61]. The choice of electrodes is critical for the success of this transformation, although it is empirical. Much lower yields were observed using other electrodes, such as Zn(+)/Al(−) and Al(+)/Al(−) (entries 11 and 12), while no product was detected using Zn(+)/Zn(−) and C(+)/C(−) (entry 13). Specifically, a cathode material with higher overpotential is typically preferred to suppress the undesired proton reduction[61]. Interestingly, a non-sacrificial anode with graphite felt (GF) was also effective in providing **3** in 75% yield (entry 14). Evaluation of different solvents uncovered that this reaction only proceeds in ethereal solvents such as $Et_2O$ (entry 1), THF (entry 15), and 1,4-dioxane (entry 16). The desired transformation was completely suppressed when swapping to non-ethereal solvents (entry 17). Additionally, increasing or decreasing the concentration turned out to be less effective (entries 18 and 19). Finally, we demonstrated the essential role of oxygen in the air by performing the reaction under nitrogen atmosphere in which styrene **1** was fully recovered (entry 20). This observation can be rationalized by the fact that β-fragmentation of the $FSO_2•$ is feasible[28,62], thus reversibly leading to the starting material styrene without enough radical trapping reagent at the cathode (e.g., $O_2$ and $FSO_2Cl$). Surprisingly, reaction without electricity also furnished the desired product **3** in 78% yield (entry 21). Presumably, an electron donor–acceptor (EDA) complex was formed between styrene **1a** and $FSO_2Cl$[63,64], thus leading to the generation of $FSO_2•$ upon daylight irradiation (see Supplementary Fig. 16). However, this EDA strategy exhibited an extremely limited styrene scope (see Supplementary Fig. 15).

### Substrate scope

With the optimized conditions in hand, we next evaluated the substrate scope of this electroreductive hydroxy fluorosulfonylation with respect to styrenes (Fig. 3a). Pleasingly, 2-, 3-, or 4-halogenated styrenes (Br, Cl, F) were well tolerated, furnishing the desired products **5–9** in 43–78% yield. Styrenes bearing electron-withdrawing groups (-$CF_3$, -CHO, -$CO_2Me$, -$NO_2$, -CN) and electron-donating groups (-Me, -OMe, -OAc, -OTs) were viable substrates, delivering β-hydroxy sulfonyl fluorides **10–21** in moderate to excellent yields. In particular, the aldehyde functionality could not be reduced under our conditions and the desired product **11** was isolated in 56%. Moreover, the sterically

**Fig. 3 | Substrate scope. a** Scope of styrenes. **b** Scope of terminal alkenes. **c** Scope of internal alkenes. **d** Site-selective hydroxyl-fluorosulfonylation. [a]conditions: **1** (0.2 mmol), **2** (2 equiv), Et$_3$SiH (2 equiv), B$_2$(OH)$_4$ (2.5 equiv), LiClO$_4$ (0.1 M), Et$_2$O (0.016 M), aluminum anode (10 mm × 15 mm × 1 mm), zinc cathode (10 mm × 15 mm × 1 mm), cell voltage ($U_{cell}$ = 8 V), undivided cell, air, rt, 16 h. Isolated yield. BRSM based on recovered starting material. [b]2 mmol scale.

hindered 2,4,6-trimethylstyrene reacted to afford difunctionalization product **18** in 49% yield. In addition, substrates bearing biphenyl, naphthyl, and benzothiophene reacted under standard conditions, furnishing the corresponding products **22**–**25** in 40–59% yield. Of note, α-methylstyrene and α-bromostyrene were successfully converted to the desired products **26** and **27** in 62% and 20% yield, respectively. Estrone and cholestanol derivatives **28** and **29** were isolated in 36% and 55% yield, respectively.

Besides styrenes, unactivated terminal alkenes were also evaluated in this hydroxy fluorosulfonylation (Fig. 3b). Terminal olefins with different long chains and branched chains were employed, giving β-hydroxy sulfonyl fluorides **30**–**34** in 40–65% yield. A variety of

functional groups, including ketone, ester, carboxylic acid, pentyl-phosphonates, and bromide, were compatible under our conditions, leading to **35**–**40** in 33–74% yield.

We examined the scope of internal alkenes next (Fig. 3c). An arrange of FSO$_2$-functionalized cycloalkanols could be accessed under mild conditions, including cyclopentanol (**41**), cyclohexanol (**42**), 1-indanol (**43**), and tetrahydronaphthalenol (**44**) from cyclic alkenes. Trisubstituted olefins such as 3-methylindene, 2-methylindene, and 4-methyl-1,2-dihydronaphthalene were transformed to sulfonyl fluorides **45**–**47** in 51–66% yield. Acyclic olefins such as β-methylstyrene or cinnamyl acetate were also effective, providing **48** and **49** in moderate yields.

Site-selective functionalization of dienes could be realized under reaction conditions (Fig. 3d). As expected, (E)-2-methyl-1-phenyl-1,3-butadiene and γ-terpinene were selectively functionalized at the less steric olefin, leading to sulfonyl fluorides **50** and **51**. Interestingly, 1,7-octadiene and 1,4-cyclohexadiene could also selectively furnish β-hydroxy sulfonyl fluorides **52** and **53** while retaining one olefin group.

**In vitro biological activities**. We also evaluated the promising bioactivities of these sulfonyl fluorides. As shown in Table 2, compounds **22**, **23**, **37**, and **45** displayed good antifungal activities against *Botrytis cinerea*, which is a serious pathogenic fungus causing severe damage to plant species worldwide[65]. Notably, **45** displayed strong inhibitory activity with $EC_{50}$ of 2.67 μg/mL, which was obviously better than chlorothalonil (see Supplementary Table 11 and Supplementary Figs. 18–20). Furthermore, several compounds exhibited significant nematicidal activity (**35**, $LC_{50} = 25.92$ μg/mL) against *Bursaphelenchus xylophilus*, which is a serious threat to pine trees and causes severe damage to forest ecosystems (see Supplementary Tables 14 and 15)[66,67]. Of note, these β-hydroxy sulfonyl fluorides typically showed improved bioactivities than β-keto sulfonyl fluorides from our previous work (see Supplementary Tables 13 and 16).

### Representative derivatizations

The synthetic utility of the hydroxyl fluorosulfonylation was further demonstrated (Fig. 4). Firstly, the electrolysis of 8 mmol of **1a** with **2** was performed under standard conditions, and the desired product **3** was obtained in 95% yield (1.55 g). Then, the alcohol moiety in β-hydroxy sulfonyl fluoride **3** could be converted efficiently to acetate or trimethyl silyl ether, resulting in **54** or **55**. In addition, oxidation of the alcohol group delivered β-keto sulfonyl fluoride **4** in 87% yield. In the presence of $AlCl_3$, 2-phenyl-2-(thiophen-2-yl) ethanesulfonyl fluoride **56** could be obtained by a Friedel−Crafts reaction of **3** with thiophene. Finally, dehydration of the alcohol in **3** was promoted by $AlCl_3$ to deliver (E)-2-phenylethene-1-sulfonyl fluoride **57**.

### Mechanistic studies

Several mechanistic experiments were conducted to gain further insight into the mechanism of this electroreductive hydroxyl fluorosulfonylation (Fig. 5a). First, a radical process was possible based on the observation that TEMPO or BHT completely inhibits the reaction. $FSO_2$ radical was also trapped by 1,1-diphenylethylene **58** with the isolation of **59** in 10% yield. Next, subjecting the observed byproduct **4** to our standard conditions did not lead to the desired product **3**, supporting that they were formed via a divergent pathway. Moreover, constant potential experiments indicated that the cathodic potential of our standard reaction was lower than −0.1 V, and a higher reductive potential led to lower reaction efficiency (Fig. 5b). Furthermore, cyclic voltammetry studies showed that $Et_3SiH$ or $B_2(OH)_4$ were not involved into the electrochemical process (see Supplementary Figs. 12 and 13).

Finally, a plausible reaction pathway for this electroreductive process is outlined in Fig. 5c, based on the abovementioned studies and our previous reports[18,19]. The reaction starts from the generation of $FSO_2$ radical via cathodic reduction of $FSO_2Cl$. Thus, the concentration of $FSO_2Cl$ in the cathode surface region is much lower than that of styrene **1a** and $O_2$. This unique feature of electrosynthesis may facilitate the reaction of benzylic radical intermediate **I**, produced from the radical addition of $FSO_2•$ to **1a**, preferentially with $O_2$ to afford benzyl peroxy radical **II**. Thus, the chlorine atom transfer with $FSO_2Cl$ can be suppressed, in contrast to previous work via photocatalysis. Subsequently, a HAT between $Et_3SiH$ and **II** delivers hydroperoxide **III**, which can be reduced by $B_2(OH)_4$ via intermediate **IV** to form borate **VI**[59,60], alone with $B(OH)_3$ detected by [11]B NMR analysis. Hydrolysis of **VI** produces the desired product **3**. Meanwhile, halogen atom transfer (XAT) of the triethylsilyl radical with $FSO_2Cl$ regenerates the $FSO_2$ radical. The constant voltage is required to improve the efficiency of this radical chain process because $Et_3Si•$ can undergo homocoupling to form hexaethyldisilane, which is confirmed by HRMS-ESI analysis. Other competing reaction pathways are also possible (Fig. 5c, right), particularly in the absence of $Et_3SiH$ and $B_2(OH)_4$.

In summary, we have realized an electroreductive hydroxyl fluorosulfonylation of alkenes that proceeds through a rapid $O_2$

### Table 2 | In vitro antifungal activities against *Botrytis cinerea*[a]

| Compound | *B. cinerea* (inhibition rate/%) | |
|---|---|---|
| | 10 μg/mL | 20 μg/mL |
| **22** | 31.9 ± 5.6 | 56.0 ± 4.4 |
| **23** | 23.4 ± 5.6 | 50.4 ± 8.8 |
| **37** | 20.6 ± 3.2 | 60.3 ± 4.4 |
| **45** | 80.8 ± 2.1 | 100.0 ± 0.0 |
| Chlorothalonil | 67.0 ± 2.1 | 81.8 ± 2.7 |

[a]Values are the mean ± standard deviation of three replicates.

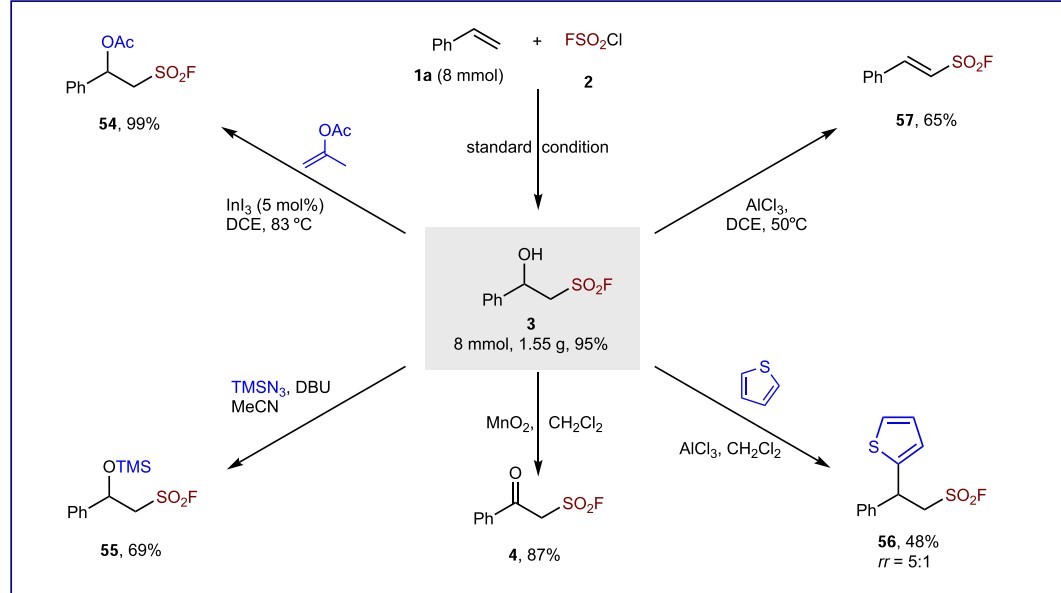

**Fig. 4 | Representative derivatizations.** DCE 1,2-dichloroethane, DBU diazabicycloundecene. For details, please see Supplementary Information (SI).

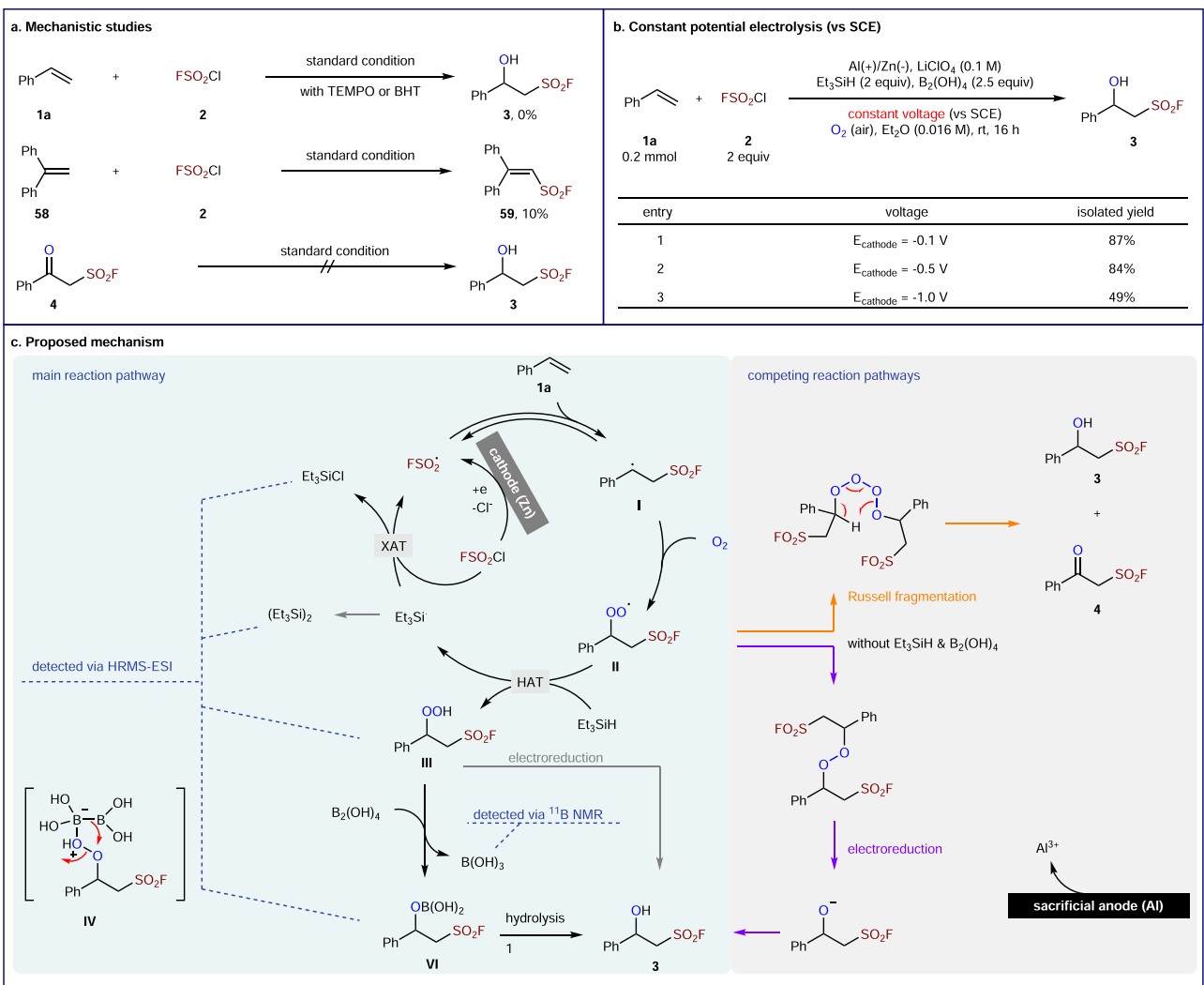

**Fig. 5 | Mechanistic experiments and proposal. a** Mechanistic studies. **b** Constant potential electrolysis. **c** Proposed mechanism.

interception by the alkyl radical and subsequent reduction of the peroxy radical. The protocol tolerates many functional groups, furnishing diverse β-hydroxy sulfonyl fluorides with potential applications for agrochemical development from alkenes under mild conditions. Furthermore, versatile follow-up derivatizations have showcased the synthetic utility to access valuable aliphatic sulfonyl fluorides, β-keto sulfonyl fluorides, and β-alkynyl sulfonyl fluorides.

## Methods

### General procedure for the synthesis of 3

A 20-mL vial with one aluminum (anode) plate electrode (10 mm × 15 mm × 1 mm), one zinc (cathode) plate electrode (10 mm × 15 mm × 1 mm) and a stir bar was charged with LiClO$_4$ (130 mg, 0.1 M), Et$_2$O (12 mL, 0.016 M), B$_2$(OH)$_4$ (0.5 mmol, 2.5 equiv), Et$_3$SiH (0.4 mmol, 2 equiv) and **1a** alkenes (0.2 mmol). Then, ClSO$_2$F was added (0.4 mmol, 2 eq, 1 M in anhydrous PhCF$_3$). The mixture was electrolyzed at a constant cell voltage of 8 V for 16 h under an atmosphere of air (1 atm, balloon). Subsequently, the reaction was quenched with water, and electrodes were rinsed with EtOAc. The resulting mixture was extracted with EtOAc and the combined organic layers were dried over Na$_2$SO$_4$ and concentrated in vacuo. The residue was purified by column chromatography to afford the desired product **3** (PE/DCM/EA = 20/20/1 ~ 5/5/1).

## Data availability

The X-ray crystallographic coordinates for structures reported in this study have been deposited at the Cambridge Crystallographic Data Centre (CCDC), under deposition number 2248411 (**46**). These data can be obtained free of charge from The Cambridge Crystallographic Data Centre via www.ccdc.cam.ac.uk/data_request/cif. The data supporting the findings of this study, including Materials and methods, optimization studies, experimental procedures, mechanistic studies, compound characterization, and NMR, are available within the article and its Supplementary Information files.

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

## Acknowledgements

Financial support provided by the Natural Science Foundation of China (32171724, S.H.) is warmly acknowledged. The authors are also thankful to Professor Fei Liu for the generous donation of *B. cinerea*, and Professor Dejun Hao for the generous donation of *B. xylophilus*. Q.F. appreciates the support from the Postgraduate Research & Practice Innovation Program of Jiangsu Province (KYCX23_1156).

## Author contributions

Q.F., T.H. and S.Q. performed the experiments. Q.F., T.H., S.Q. and S.H. analyzed the data. S.H. designed and directed the project and wrote the paper with feedback from P.X. and S.L.

## Competing interests

The authors declare no competing interest.
