## [Peer Review File · Nature Communications]

REVIEWER COMMENTS

Reviewer #1 (Remarks to the Author):

This manuscript by Huang and co-workers described the electrochemical hydroxyl fluorosulfonylation of alkenes with sulfonyl chlorofluoride and air. This mild protocol displays excellent functional group compatibility and broad scope, and provides a convenient access to diverse β -hydroxy sulfonyl fluorides. As the β -hydroxy sulfonyl fluorides are very useful synthetic building blocks, this work will be interest to a number pf organic and medicinal chemists. This manuscript was recommended for publication in Natural Communication when the following major modification was made.

1) The introduuction part of this manuscript was not organized well. Please ask a chemist who is native English speaker to re-written this part.

2)Fig. 2: What is formed from the reduction of compound 4 ?

3) Regarding to the optimization the reaction conditions (Table 1):

a) please explain why electrodes played a key role for the formation of the desired product 3 in good yield.

b) It was very difficult to understand that styrene was fully recovered underr nitrogen atmosphere. From the proposed reaction mechanism, β -chloro sulfonyl fluoride should be formed.

4) Regarding to the reaction mechanism (Figure 5): Et_3SiH and eespecially $\text{B}_2(\text{OH})_4$ are very important for the fromation of β -hydroxy sulfonyl fluorides. But 40% of the desired product was obtained without Et_3SiH and $\text{B}_2(\text{OH})_4$. Please rationalize these results.

Reviewer #2 (Remarks to the Author):

The presented work shows an interesting difunctionalization of alkenes, in principle, utilizing electroreductive conditions. Products were tested against *Botrytis cinerea* and successfully derivatized showing the potential utility of the procedure. However, there are critical points to discuss that will be essential for the decision of this reviewer about the suitability for publication in Nature Communications:

1. During the control experiments, the absence of electricity provides quite a good yield (78%, page S6). The title of the manuscript says “electroreductive...”, but this result shows that reaction proceeds in the absence of electrical current. In the main text, the only mention to control experiments is “Control experiments showed that Et₃SiH and B₂(OH)₄ both were important (entries 3&4)”. Authors should explain this result. This is an inflection point that might change the entire discussion and conclusions of the work.
2. Authors should propose electricity-free mechanism for the reaction.
3. How many F/mol were passed through the reaction? This might help to understand the control experiment result.
4. Why was the reaction set at constant voltage instead of constant current?

Reviewer #3 (Remarks to the Author):

This manuscript describes a new hydroxy fluorosulfonylation reaction of alkenes using an electroreductive strategy. The manuscript begins with a thorough introduction that covers all of the key references and places the current work in the appropriate context. The optimization section is interesting, but more detail should be included about the key reaction conditions (i.e. the Et₃SiH and the B₂(OH)₄) as well as a bit more background on these reagents. The reaction scope presented in Figure 3 is extensive. I appreciate that the authors included a full scope, including substrates that do not perform very well under the reaction conditions. There are a sufficient number of substrates that are successful that indicate that the reaction has a reasonable scope. The antifungal activity is interesting, but is not very informative about the specific activity of the beta-hydroxy sulfonyl fluorides (i.e. is this subunit critical for the activity or will other sulfonyl fluorides provide comparable activity). The derivatization section is useful as are the mechanistic studies. The proposed mechanism is consistent with the evidence provided.

Overall, this manuscript is very well written and presented. The supporting information is clear and the presentation is up to (and exceeds) the standards of the field. The only clarifying information that should be included is the diastereoselectivities of some of the substrates (such as 41 and 43) in the main manuscript. The question of publication in Nature Communications comes down to the novelty of the method. It is true that these motifs have not been synthesized very often, but that is the case of many functional group pairings. Furthermore, the authors have not made a very strong case regarding why these should be made. The antifungal activities may be interesting, but they do not make a strong case for this specific motif. The core electrochemical reactivity has already been demonstrated by the same group. The novelty of this manuscript relies on the reductive scission of the hydroperoxide, which also was not emphasized/stressed in this manuscript. While the results are

interesting and the manuscript is well-prepared, the novelty does not reach the level of that expected in a journal of this caliber. I recommend that it is published in a more specialized journal with (very) minor revisions.

Point-to-Point Response:

Reviewer #1 (Remarks to the Author):

Comments:

This manuscript by Huang and co-workers described the electrochemical hydroxyl fluorosulfonylation of alkenes with sulfonyl chlorofluoride and air. This mild protocol displays excellent functional group compatibility and broad scope, and provides a convenient access to diverse β -hydroxy sulfonyl fluorides. As the β -hydroxy sulfonyl fluorides are very useful synthetic building blocks, this work will be interest to a number pf organic and medicinal chemists. This manuscript was recommended for publication in Natural Communication when the following major modification was made.

Response: We thank the reviewer for the positive comments and helpful suggestions, which helped significantly to improve our manuscript.

1) The introduction part of this manuscript was not organized well. Please ask a chemist who is native English speaker to re-written this part.

Response: As suggested, we have redrawn Figure 1 and rewritten the introduction, particularly considering the following suggestions. The updated introduction makes the following claims:

a) The incorporation of a hydroxyl group can significantly change binding affinity and pharmacokinetic properties of drug molecules (J. Med. Chem. 2019, 62, 8915;). As such, we envisioned that the combined β -hydroxy sulfonyl fluoride motif might exhibit improved bioactivity compared to previously reported β -keto sulfonyl fluorides (Fig. 1a, I).

b) It is challenging to construct β -hydroxy sulfonyl fluoride motif. The direct reduction of the ketone group in β -keto sulfonyl fluorides was failed. In addition, identification of a suitable reduction system to suppress the competing pathway including Russell fragmentation, hydrogen atom transfer (HAT), and carbon anion generation.

2) Fig. 2: What is formed from the reduction of compound 4?

Response: The major byproduct is acetophenone. Presumably, the FSO_2 group works as a good leaving group and is substituted by hydride. We updated this information in the introduction and Figure 1.

3) Regarding to the optimization the reaction conditions (Table 1):

a) please explain why electrodes played a key role for the formation of the desired product 3 in good

yield.

Response: It is known that the electrode material can significantly influence electron transfer (Angew. Chem. Int. Ed. 2020, 59, 18866). The electron transfer takes place on the surface of the electrodes. The choice of electrodes is critical for the success of this transformation, although it is empirical. Specifically, a cathode material with higher overpotential is typically preferred to suppress the undesired proton reduction for our reaction. We added this discussion to the manuscript (optimization).

b) It was very difficult to understand that styrene was fully recovered under nitrogen atmosphere.

From the proposed reaction mechanism, β -chloro sulfonyl fluoride should be formed.

Response: We regret that the manuscript did not convey its message clearly. This observation can be rationalized by the fact that β -fragmentation of the $\text{FSO}_2\cdot$ is feasible (Chem. Sci. 2021, 12, 11762; Angew. Chem. Int. Ed. 2022, 61, e202115593), thus reversibly leading to the starting material styrene without enough radical trapping reagent at the cathode (e.g., O_2 and FSO_2Cl). The consumption of FSO_2Cl at the cathode results in a low local concentration of FSO_2Cl . We emphasized this point in the manuscript and updated in the mechanism pathway (Figure 5).

4) Regarding to the reaction mechanism (Figure 5): Et_3SiH and especially $\text{B}_2(\text{OH})_4$ are very important for the formation of β -hydroxy sulfonyl fluorides. But 40% of the desired product was obtained without Et_3SiH and $\text{B}_2(\text{OH})_4$. Please rationalize these results.

Response: Other competing reaction pathways are also possible (Fig. 5c, right), particularly in the absence of Et_3SiH and $\text{B}_2(\text{OH})_4$. We emphasized this point in the manuscript and updated the mechanism pathway in Figure 5. For example, Russell fragmentation of alkyl peroxy radical could provide alcohol product **3** and ketone **4** (J. Am. Chem. Soc. 1957, 79, 3871; J. Am. Chem. Soc. 1968, 90, 1056).

Reviewer #2 (Remarks to the Author):

Comments:

The presented work shows an interesting difunctionalization of alkenes, in principle, utilizing electroreductive conditions. Products were tested against *Botrytis cinerea* and successfully derivatized showing the potential utility of the procedure. However, there are critical points to discuss that will be essential for the decision of this reviewer about the suitability for publication in Nature Communications:

Response: We thank the reviewer for the positive comments and helpful suggestions, which helped significantly to improve our manuscript.

1. During the control experiments, the absence of electricity provides quite a good yield (78%, page S6). The title of the manuscript says “electroreductive...”, but this result shows that reaction proceeds in the absence of electrical current. In the main text, the only mention to control experiments is “Control experiments showed that Et_3SiH and $\text{B}_2(\text{OH})_4$ both were important (entries 3&4)”. Authors should explain this result. This is an inflection point that might change the entire discussion and conclusions of the work.

Response: We regret that the manuscript did not convey its message clearly. Presumably, an electron donor–acceptor (EDA) complex was formed between styrene **1a** and FSO_2Cl , thus leading to the generation of $\text{FSO}_2\cdot$ upon daylight irradiation. Preliminary experiments were conducted to confirm this mechanism (see Figure S16 in the Supplementary Information). We have reported that FSO_2Cl could interact with some specific electron-rich substrates to form an EDA complex (Org. Lett. 2023, 25, 3109; Org. Chem. Front., 2023, 10, 3805).

Moreover, we have examined the scope of this EDA approach. However, this EDA strategy exhibited an extremely limited styrene scope (see Figure S15 in the Supplementary Information). We can only realize this transformation with good scope under electroreductive conditions.

2. Authors should propose electricity-free mechanism for the reaction.

Response: As suggested, we added the mechanism via EDA photoactivation in the Supplementary Information (Figure S16 in the Supplementary Information).

3. How many F/mol were passed through the reaction? This might help to understand the control experiment result.

Response: We included this information at the footnote of Table 1, and 0.89 F/mol were passed through the reaction. To make it more understandable, we also updated in the reaction equation in Table 1. Therefore, we proposed a radical chain process. The control experiments also support this.

4. Why was the reaction set at constant voltage instead of constant current?

Response: When constant current conditions were employed, the potential gradually increased over the reaction time and more byproducts were formed. As such, constant cell voltage conditions were employed to avoid undesired redox-processes. We updated this information in the manuscript.

Reviewer #3 (Remarks to the Author):

Comments:

This manuscript describes a new hydroxy fluorosulfonylation reaction of alkenes using an electroreductive strategy. The manuscript begins with a thorough introduction that covers all of the key references and places the current work in the appropriate context. The optimization section is interesting, but more detail should be included about the key reaction conditions (i.e. the Et₃SiH and the B₂(OH)₄) as well as a bit more background on these reagents. The reaction scope presented in Figure 3 is extensive. I appreciate that the authors included a full scope, including substrates that do not perform very well under the reaction conditions. There are a sufficient number of substrates that are successful that indicate that the reaction has a reasonable scope. The antifungal activity is interesting, but is not very informative about the specific activity of the beta-hydroxy sulfonyl fluorides (i.e. is this subunit critical for the activity or will other sulfonyl fluorides provide comparable activity). The derivatization section is useful as are the mechanistic studies. The proposed mechanism is consistent with the evidence provided.

Overall, this manuscript is very well written and presented. The supporting information is clear and the presentation is up to (and exceeds) the standards of the field. The only clarifying information that should be included is the diastereoselectivities of some of the substrates (such as 41 and 43) in the main manuscript. The question of publication in Nature Communications comes down to the novelty of the method. It is true that these motifs have not been synthesized very often, but that is the case of many functional group pairings. Furthermore, the authors have not made a very strong case regarding why these should be made. The antifungal activities may be interesting, but they do not make a strong case for this specific motif. The core electrochemical reactivity has already been demonstrated by the same group. The novelty of this manuscript relies on the reductive scission of the hydroperoxide, which also was not emphasized/stressed in this manuscript. While the results are interesting and the manuscript is well-prepared, the novelty does not reach the level of that expected in a journal of this caliber. I recommend that it is published in a more specialized journal with (very) minor revisions.

Response: We thank the reviewer for the detailed comments, which helped to significantly improve

the manuscript.

1. We have redrawn Figure 1 and rewritten the introduction, particularly considering all the above suggestions. The updated introduction makes the following claims:

a) The incorporation of a hydroxyl group can significantly change the binding affinity and pharmacokinetic properties of drug molecules. As such, we envisioned that the combined β -hydroxy sulfonyl fluoride motif might exhibit improved bioactivity compared to previously reported β -keto sulfonyl fluorides (Fig. 1a, I). Indeed, these β -hydroxy sulfonyl fluorides typically showed improved bioactivities than β -keto sulfonyl fluorides from our previous work (See Tables S4&S7 in the Supplementary Information).

b) It is challenging to construct β -hydroxy sulfonyl fluoride motif. The direct reduction of the ketone group in β -keto sulfonyl fluorides was failed. In addition, identification of a suitable reduction system would be key to suppress the competing pathway including Russell fragmentation, hydrogen atom transfer (HAT), and carbon anion generation. That means that a suitable reduction system only works for the scission of the hydroperoxide to provide desired β -hydroxy sulfonyl fluorides selectively.

2. More details were included in the optimization section (Figure 2, Table 1, and the corresponding main text).

a) When constant current conditions were employed, the potential gradually increased over the reaction time and more byproducts were formed. As such, constant cell voltage conditions were employed to avoid undesired redox-processes.

b) Inspired by Mukaiyama hydration, various hydride donors (silanes) were examined to suppress ketone formation. That is how we get to Et_3SiH .

c) Since $\text{B}_2(\text{OH})_4$ could be used a deoxygenating agent and boronic acids were able to reduce peroxides (Synlett 2013, 24, 2695; Tetrahedron Lett. 2017, 58, 4572.), here $\text{B}_2(\text{OH})_4$ likely acted as a reducing agent for the reduction of the hydroperoxide intermediate.

d) It is known that the electrode material can significantly influence the electron transfer (Angew. Chem. Int. Ed. 2020, 59, 18866). The electron transfer takes place on the surface of the electrodes. The choice of electrodes is critical for the success of this transformation, although it is empirical. Specifically, a cathode material with higher overpotential is typically preferred to suppress the undesired proton reduction for our reaction.

e) Styrene was fully recovered under nitrogen atmosphere. This observation can be rationalized by the fact that β -fragmentation of the $\text{FSO}_2\cdot$ is feasible (Chem. Sci. 2021, 12, 11762; Angew. Chem. Int. Ed. 2022, 61, e202115593), thus reversibly leading to the starting material styrene without enough radical trapping reagent at the cathode (e.g., O_2 and FSO_2Cl). The consumption of FSO_2Cl at the cathode results in a low local concentration of FSO_2Cl .

f) Surprisingly, reaction without electricity also furnished the desired product 3 in 78% yield (entry

21). Presumably, an electron donor–acceptor (EDA) complex was formed between styrene **1a** and FSO₂Cl, thus leading to the generation of FSO₂• upon daylight irradiation. Preliminary experiments were conducted to confirm this mechanism (see Figure S16 in the Supplementary Information). We have reported that FSO₂Cl could interact with some specific electron-rich substrates to form an EDA complex (Org. Lett. 2023, 25, 3109; Org. Chem. Front., 2023, 10, 3805). Moreover, we have examined the scope of this EDA approach. However, this EDA strategy exhibited an extremely limited styrene scope (see Figure S15 in the Supplementary Information). We can only realize this transformation with good scope under electroreductive conditions.

3. The diastereoselectivities of some of the substrates (such as **41** and **43**) were updated.

4. We emphasized how we suppress the competing pathway in the manuscript and updated the mechanism pathway in Figure 5.

The β-hydroxy sulfonyl fluoride motif does show potential applications for agrochemical development and is easily converted to valuable aliphatic sulfonyl fluorides, β-keto sulfonyl fluorides, and β-alkynyl sulfonyl fluorides. Moreover, β-hydroxy sulfonyl fluorides cannot be synthesized either from alkynes under the current conditions or alkenes under our previous conditions. These findings also better differentiate current work from our prior report.

We addressed all questions and concerns of the Reviewer in this substantial revision. We believe all outstanding issues have been resolved and hope the Reviewer can now recommend the manuscript for publication in Nature Communications.

We thank all the reviewers for insightful suggestions and comments!

REVIEWERS' COMMENTS

Reviewer #1 (Remarks to the Author):

My comments were fully addressed in this revised manuscript. Of course, this revised manuscript was recommended for publication in Nature Communication.

Reviewer #2 (Remarks to the Author):

All concerns were answered, so I recommend its publication in Nature Communications.